# Chronic Enteropathy and Vitamins in Dogs

**DOI:** 10.3390/ani15050649

**Published:** 2025-02-23

**Authors:** Yu Tamura

**Affiliations:** 1Nagaya Animal Medical Center, Nagoya 468-0024, Aichi, Japan; yu.tamura.dvm.phd@gmail.com or tamura@azabu-u.ac.jp; 2Collaborate Research Worker, Laboratory of Small Animal Internal Medicine, Azabu University, 1-17-71 Fuchinobe, Chuo-ku, Sagamihara 252-5201, Kanagawa, Japan

**Keywords:** canine, cobalamin, chronic enteropathy, treatment, vitamin D

## Abstract

Gastrointestinal signs are commonly encountered in dogs. Transient acute gastroenteritis improves quickly with symptomatic treatment; however, if symptoms persist for more than 3 weeks, chronic enteropathy (CE) should also be considered. Dietary therapy is the first step in the treatment of chronic enteropathy. Some vitamins, which are poor prognostic factors, should be supplemented. In this review, the author outlines the therapeutic management of CE and describes vitamins B_12_ and D, which are considered particularly important.

## 1. Introduction

Chronic enteropathy (CE) or chronic inflammatory enteropathy is a group of diseases with multiple and different etiologies characterized by chronic gastrointestinal signs such as vomiting, diarrhea, anorexia, weight loss for more than 3 weeks, and inflammatory cell infiltration, such as lymphoplasmacytic cells in the intestinal mucosal lamina propria [1,2,3]. The diagnosis further requires ruling out other diseases, such as parasitic infections, tumors, pancreatitis, exocrine pancreatic insufficiency, metabolic diseases, and endocrine diseases, such as hypoadrenocorticism. The disease is classified as food-responsive enteropathy (FRE), which responds to dietary therapy; immunosuppressant-responsive enteropathy (IRE), which responds to immunosuppressive therapy such as steroids; and non-responsive enteropathy (NRE), which does not respond to these therapies [1]. IRE is known as inflammatory bowel disease (IBD) [3]; however, since IBD is derived from the name of a human disease and its pathogenesis is completely different, IRE is now expected to become the predominant name in dogs [1]. Antibiotic-responsive enteropathy (ARE) was previously thought to be a disease that responds to antimicrobial agents such as metronidazole and tylosin [1]; however, it is now known to cause dysbiosis in the intestinal microbiota. Therefore, it is now recommended that antimicrobials should be used only when infection is suspected [4,5]. Moreover, ARE has been proposed to replace microbiota-related modulation-responsive enteropathy (MrMRE) [2].

### 1.1. Food-Responsive Enteropathy (FRE) and Its Therapy

It has been reported that FRE responds to an elimination diet as early as a few days or as late as 10–14 days [1,6]. It is difficult to differentiate CE using methods other than treatment response. It has been reported that 79% of cases that went into remission on an elimination diet did not relapse when returned to the previous original diet after 14 weeks [6], and 21% of FRE cases that relapsed after returning to the previous diet were thought to be food allergies [6,7]. FRE accounts for approximately 65% of CE [8,9] in young dogs (median age of onset: 3.4–5.3 years [6,9]), and many cases have a good prognosis [6].

However, some dogs with pruritus, which is thought to be caused by food allergies, showed recurring clinical signs and were euthanized due to poor responses to treatment [6]. Pruritus, which was not included in the previous clinical score [the canine inflammatory bowel disease activity index (CIBDAI)] [10], was included in the later clinical score [the canine chronic enteropathy clinical activity index (CCECAI)] [6] (Table 1). Therefore, in cases of CE complicated by pruritus, a food allergy might be suspected, and the possibility of a poor response to treatment might be assumed.

Recently, a report on the relationship between house dust mites and CE was published [11], and it is expected that more research will be conducted on environmental allergies other than food allergies.

It has been reported that 70% of FRE cases are colonic diarrhea [6], and small bowel lesions (small bowel diarrhea) have more severe clinical signs and a worse response to treatment than colonic lesions (large bowel diarrhea) [6,9]. In other words, colonic diarrhea is more likely to have milder clinical signs and a better response to treatment than small bowel diarrhea.

Chronic idiopathic large bowel diarrhea (CILBD) is a differential diagnosis of chronic diarrhea of unknown cause [12,13]. CILBD is sometimes called fiber-responsive colonic diarrhea because many cases respond to dietary fiber. The prognosis of CILBD is good, and most cases respond to the addition of dietary fiber [12,13]. As a treatment, half a teaspoon of psyllium is added to food in toy breeds, one teaspoon in small dogs, two teaspoons in medium-sized dogs, and three teaspoons in large dogs; a median amount of 1.3 g/kg/day has been reported [12]. A high-fiber diet containing 7–9% or more dietary fiber in dry matter is also useful [13]. Therefore, it is worthwhile to try psyllium and high-fiber diets before using elimination diets in cases of chronic colonic diarrhea.

The prognosis in dogs with FRE is good, with 97% (38/39) of cases surviving for more than 3 years [6].

### 1.2. Antibiotic-Responsive Enteropathy (ARE) or Microbiota-Related Modulation-Responsive Enteropathy (MrMRE) and Its Therapy

In the past, ARE was diagnosed when dogs responded to experimental treatment with antimicrobial agents (especially metronidazole) as the next step after trying several elimination diets for 2 weeks [1,14]. Recently, however, the increase in the number of drug-resistant bacteria has become a worldwide problem, and careful use of antimicrobial agents has been proposed for the diagnosis of CE [4]. Specifically, endoscopic biopsy without the use of antimicrobial agents is recommended in cases of chronic diarrhea that do not respond to elimination diets and prebiotics and/or probiotics [4,5]. Therefore, antimicrobial agents should be used only when histopathological results show bacterial infections [4]. Furthermore, there is some evidence in previous reports that tylosin (5–25 mg/kg PO q24h) is the only effective antimicrobial against chronic diarrhea [1,15,16,17]. However, the world standard is that antimicrobial agents should not be used in dogs that do not need them, because these reports did not exclude FRE. Currently, the only diseases that require the use of antimicrobial agents after endoscopy are infections and granulomatous colitis [4,18]. Granulomatous colitis is a common disease caused by *E. coli* in Boxers and French Bulldogs [19,20]. The first-line drug is reported as enrofloxacin, but it is generally a second-line drug, and resistant strains have been reported in recent years [19,20]. Therefore, treatment with appropriate antimicrobial agents is necessary to perform bacterial cultures and drug susceptibility tests.

Recently, ARE was proposed to be replaced with microbiota-related modulation-responsive enteropathy (MrMRE) [2]. This category has been proposed, including enteropathy in response to pre/pro/symbiotic products or fecal microbiota transplantation, although there is not yet a reliable diagnostic criterion for the MrMRE category [2].

### 1.3. Immunosuppressant-Responsive Enteropathy (IRE) and Its Therapy

Prednisolone/prednisone (PSL) was the first choice of treatment for IRE (Table 2). Prednisone is metabolized in the liver to act as prednisolone [21]. Allenspach et al. reported that 10 of 21 dogs treated with PSL remained relapse-free for three years [6]. Eight of the eleven dogs that did not respond to PSL were treated with cyclosporine, and three were euthanized [6]. Garcia-Sancho et al. reported that all 16 dogs treated with PSL showed a response to treatment during the first 90 days after initiation, whereas the pathological results did not change before or after treatment [22]. Jergens et al. compared the efficacy of PSL alone or PSL plus metronidazole and found that 22 of 25 dogs treated with PSL alone for 3 weeks responded to treatment and 24 of 29 dogs treated with PSL plus metronidazole responded to treatment, indicating no difference in response to treatment with metronidazole [23]. Similarly, Heilmann et al. reported that when 34 dogs were treated with PSL alone or PSL plus metronidazole for 3 weeks, only 3 did not respond to treatment [24]. Dye et al. compared the efficacy of PSL and budesonide for 6 weeks; 11 of 16 dogs treated with PSL showed treatment response, and 14 of 18 dogs treated with budesonide showed treatment response, indicating no difference in treatment efficacy between PSL and budesonide [25]. White et al. compared the efficacy of PSL alone and PSL plus probiotics for 8 weeks; 10 of 12 dogs with PSL alone responded to treatment, and 12 of 14 dogs with PSL plus probiotics responded to treatment, showing no difference in the efficacy of treatment with probiotics but a difference in the expression of tight junctions, which are mucosal barriers [26]. In conclusion, although there are some differences among reports, a short-term therapeutic effect can be expected with the use of PSL. The median survival time (MST) is reported to be >1300 days in dogs with CE [27].

Budesonide is a glucocorticoid that is metabolized by a first-pass effect in the liver, making it less likely to cause systemic side effects than other glucocorticoids [28]. Therefore, it is sometimes used when the systemic side effects of PSL are unacceptable. While there are reports of efficacy as mentioned above [25,29], others have reported no efficacy compared to placebo owing to severe clinical signs [30]; thus, the results are controversial. It is worth using budesonide for dogs with IRE that did not respond to PSL; however, further studies are needed because the dosage of budesonide varies among reports.

**Table 2 animals-15-00649-t002:** Reports of immunosuppressive drugs used in dogs with IRE.

Drug	Initial Dose	Method	Response Rate	Reference
Prednisolone	2 mg/kg q24h	tapered off over 10 weeks	48% (10/21)	Allenspach et al., 2007 [6]
Prednisone	1 mg/kg q12h	tapered off over 90 days until 0.5 mg/kg q48h	100% (16/16)	García-Sancho et al., 2007 [22]
	1 mg/kg q24h	continued for 3 weeks	85% (46/54)	Jergens et al., 2010 [10]
	1 mg/kg q12h	continued for 3 weeks	91% (31/34)	Heilmann et al., 2012 [24]
	1 mg/kg q12h	continued for 3 weeks then reduced by half and continued for 3 weeks	69% (11/16)	Dye et al., 2013 [25]
	0.5–1 mg/kg q12h	continued for 3 weeks then reduced to 0.5 mg/kg q12h over 2 weeks	85% (22/26)	White et al., 2017 [26]
Budesonide	0.1–0.3 mg/kg q24h	continued for 6 weeks	78% (14/18)	Dye et al., 2013 [25]
	3 mg/m^2^ q24h	continued for 30 days	73% (8/11)	Pietra et al., 2013 [29]
	3 mg/head q24h	continued for 30 days	0% (0/7)	Rychlik et al., 2016 [30]
Cyclosporine	5 mg/kg q24h	continued for 10 days	86% (12/14)	Allenspach et al., 2006 [31]
	5 mg/kg q24h	continued for 10 days	25% (2/8)	Allenspach et al., 2007 [6]

IRE, immunosuppressant-responsive enteropathy.

Cyclosporine has been reported to be an immunosuppressive agent for the treatment of dogs with IRE. In one report, 8 of 11 dogs that did not respond to PSL were treated with cyclosporine; however, 6 dogs were euthanized because of poor response [6]. In contrast, 12 of 14 dogs treated with cyclosporine for similar PSL-resistant cases responded to treatment [31]. The long-term prognosis after a 10-week observation period in these twelve dogs was that although one dog was euthanized, eight dogs were able to withdraw cyclosporine and remained in remission for two years, and the remaining three dogs sustained remission for three years with continued cyclosporine [31]. The results vary widely among reports, but cyclosporine may be used in PSL-resistant cases. Although azathioprine, mycophenolate mofetil, and chlorambucil have been described as immunosuppressive agents [1], no evidence of efficacy has been reported to date.

### 1.4. Non-Responsive Enteropathy (NRE) and Its Therapy

T-cell gastrointestinal small cell lymphoma (GISCL) is considered to occur in dogs with NRE. GISCL are mainly treated with PSL plus chlorambucil, and four previous papers have reported MSTs of 424–700 days [27,32,33,34]. The poor prognosis of Shiba dogs with CE is well known in Japan [35,36,37] and is more pronounced in cases with clonality-positive results on polymerase chain reactions for antigen receptor rearrangement (PARR) analysis (median survival is reported to be 48 days for clonality-positive cases and 271 days for clonality-negative cases) [36]. In addition, the T-cell receptor γ gene showed monoclonal proliferation (clonality positive) in most cases with a PARR analysis in Shiba dogs with CE, and the possibility of GISCL was pointed out [36]; later, it was reported that the incidence of T-cell GISCL is higher in Shiba dogs [38]. Since the age of onset of CE and GISCL tends to be younger in Shiba dogs than in other breeds [35,38], an earlier transition from enteritis to lymphoma may contribute to poor prognosis. Although breed-specific reasons are not clear, Shiba dogs are also a predominant breed for atopic dermatitis, and in recent years, dysbiosis of the gut and skin microbiota has been reported in Shiba dogs with atopic dermatitis [39]. Therefore, it is conceivable that the intestinal microbiota is associated with immunological abnormalities related to tumorigenesis.

### 1.5. Protein-Losing Enteropathy (PLE) and Its Therapy

Protein-losing enteropathy (PLE) is a group of diseases associated with hypoproteinemia due to gastrointestinal disorders including CE [1]. PLE caused by FRE responds better to a low-fat diet (LFD) than to an elimination diet [1], which is especially true for Yorkshire terriers with PLE [40,41]. In addition, a homemade ultra-low-fat diet (ULFD) is used for PLE caused by primary intestinal lymphangiectasia (IL) [42,43]. Specifically, the maintenance energy requirement per body weight was calculated and used to calculate the food requirement in the ratio of chicken breast to potato/ rice = 1:2 [42]. It is also possible for owners to calculate the calory requirements using the Food Composition Database [https://fooddb.mext.go.jp/ (accessed on 28 December 2024); for example, chicken breast: 1.34 kcal/g, potato: 0.83 kcal/g, rice: 1.68 kcal/g]. However, the ULFD is not a complete nutritional diet. Although ULFD is effective in the treatment of IL in the short term [42,43], it is reported that the development of hypocalcemia is caused by nutritional secondary hyperparathyroidism when using ULFD for the long term in dogs with PLE [44]. Therefore, LFD was added gradually to ULFD after the improvement in albumin concentrations in a retrospective study of ULFD treatment in dogs with PLE [43]. Thus, when ULFD is administered to dogs with PLE for a long-term period, albumin and calcium concentrations should be monitored and, whenever possible, a complete nutritional diet should be used at a rate of at least 70%. In addition, it has been reported that cases of PLE that did not respond to diet were not primary IL, but secondary IL caused by lymphoplasmacytic enteritis [43]; therefore, in cases of PLE that are not a primary IL, it might be difficult to control the clinical signs with dietary therapy alone.

### 1.6. Prognostic Factors in Dogs with CE and PLE

Hypoalbuminemia [6], hypocobalaminemia [6], CIBDAI/CCECAI score [6], severity of pathology [45], and Shiba dogs [35,36,37,46] have been reported as prognostic factors for dogs with CE. Specifically, a serum albumin concentration of <2.0 g/dL, serum cobalamin concentration of <200 ng/dL, CIBDAI score of >9, and CCECAI score of >12 are reported as poor prognostic factors [6]. Regarding the severity of the pathology, MST was >800 days for mild/moderate disease and 542 days for severe disease, with a significant difference [45].

Prognostic factors for PLE were reported to be elevated CIBDAI/CCECAI scores [47,48], clonality-positive PARR results [47], an elevated BUN [47,48,49], high body weight [49], hypoproteinemia and hypocholesterolemia, a CCECAI score >5 after 1 month of treatment [49], hypovitaminosis D [50], and dilated small bowel findings on an abdominal ultrasound [51]. Cases that could be controlled by dietary therapy alone and those that could be managed with a feeding tube were reported to have a good prognosis [52]. Moreover, an increase in serum CRP concentration after 1–3 days of in-hospital treatment was reported as a negative risk factor, and a higher mortality was identified in Pugs [53]. However, when comparing these results, it is important to note that the inclusion criteria for PLE in foreign reports are CE [48,49,50,52,53], whereas those for PLE in Japanese reports include GISCL and gastrointestinal large cell lymphoma (GILCL) [47,51]. Because the prognosis of especially GILCL is clearly worse than that of CE [45,47], caution should be used when comparing these results.

In summary, cobalamin (vitamin B_12_) and vitamin D are important prognostic vitamins for CE and PLE.

### 1.7. Vitamin B_12_ in Dogs with CE

Vitamin B_12_ is a water-soluble vitamin, of which cobalamin is the biologically active form [54]. Cobalamin is absorbed from food via receptors (Figure 1). Food-derived cobalamin complexes are released by pepsinogen and gastric acid in the stomach. The released cobalamin is immediately bound to R-proteins derived from gastric juice and saliva and transported to the duodenum. In the duodenum, R protein is degraded by pancreatic proteases, and cobalamin is bound to intrinsic factors (IF). In dogs, IF is produced in the stomach and pancreas [55]. On the other hand, IF is produced mainly by the gastric mucosa in humans, whereas it is synthesized exclusively by the exocrine pancreas in domestic cats [55,56]. The cobalamin and IF complex are absorbed via receptors in the ileum and bound to transcobalamin, another binding protein that is transported to tissues for utilization [57]. Cobalamin is known to play an important role as a cofactor for various enzymes, among which methionine synthase and methylmalonyl-CoA mutase are essential for the synthesis of methionine in the cytosol and for reactions in the tricarboxylic acid cycle associated with the synthesis of succinyl-CoA in mitochondria (Figure 2) [58,59,60]. When these metabolisms are disrupted by cobalamin deficiency, methylmalonic acid (MMA) and homocysteine accumulate as different end products (Figure 2). In humans, pernicious anemia, neuropathy, MMA-urea, atherosclerosis, and cerebrovascular disease are associated with cobalamin deficiency [61]. Reasons for cobalamin deficiency include congenital defects in receptor function, excessive competition with intestinal bacteria, and decreased mucosal absorption capacity, of which decreased mucosal absorption capacity is the most important factor [54].

Hypocobalaminemia is commonly observed in dogs with CE [6,62,63]. Since cobalamin deficiency of dietary origin has not been reported to date, cobalamin deficiency in dogs with CE may be attributed to the absorption process [54]. Theoretically, there are three main mechanisms that reduce cobalamin utilization in the small intestine: congenital impairment of receptor function, reduced mucosal absorption capacity, and excessive competition with the intestinal microflora [54]. Of these three mechanisms, decreased mucosal absorption capacity due to mucosal inflammation may be the most important in dogs with CE.

Hypocobalaminemia is a poor prognostic factor for dogs with CE and thus requires supplementation [6]. Indeed, cobalamin supplementation has been reported to improve clinical signs in dogs with CE and hypocobalaminemia [62]. Cobalamin supplementation has been administered subcutaneously to account for the decreased absorption in the intestinal mucosa [54]. The usual method has been to administer weekly subcutaneous (SC) injections for six consecutive weeks followed by additional weekly to monthly amounts based on serum cobalamin concentrations if necessary [54]. Recently, however, it has been reported that an oral (PO) administration of cobalamin has the same effect as SC injection in dogs with CE [59,62,63]; therefore, PO administration might be more beneficial considering homecare for outpatients. The SC supplemented amounts are reported as 250–1200 µg of hydroxocobalamin depending on body weight [54], and the PO feeding amounts are reported in dogs with body weights of 1–10 kg that received 250 µg, 10–20 kg that received 500 µg, and >20 kg that received 1000 µg daily [59,63] (Table 3). It has also been confirmed that serum cobalamin concentrations increase with cobalamin-containing vitamin B combination tablets in Japan [64].

### 1.8. Vitamin D in Dogs with PLE

It is well known that vitamin D is important for bone metabolism and calcium homeostasis. However, since the discovery of vitamin D receptors in various immune cells in humans, it has been considered that vitamin D may affect approximately 2000 genes, equivalent to 10% of the human genome, and its multifaceted effects beyond bone metabolism and calcium regulation have attracted attention [65,66]. Unlike humans, dogs and cats cannot synthesize vitamin D in the skin through sunlight and therefore depend on food-derived intake [67,68]. There are two types of vitamin D in nature: plant-derived ergocalciferol (vitamin D_2_) and animal-derived cholecalciferol (vitamin D_3_). Natural vitamin D is a generic term for both [68]. When natural vitamin D is ingested and absorbed from the intestinal tract, it is transported to the liver by vitamin D-binding protein (VDBP), where it undergoes hydroxylation at carbon 25 to form 25-hydroxyvitamin D [25(OH)D] [68]. Owing to the differences in vitamin D metabolism, carnivorous cats cannot synthesize 25(OH)D from ergocalciferol, unlike omnivorous dogs [69,70]. Subsequently, 25(OH)D is transported to the kidney by VDBP and undergoes hydroxylation at carbon 1 in the proximal tubule to form 1,25-dihydroxyvitamin D [1,25(OH)_2_D] [68] (Figure 3). 1,25(OH)_2_D, also called calcitriol, is the major active form of vitamin D and is known to exert various effects via the vitamin D receptor in vivo [68]. The synthesis of 1,25(OH)_2_D, the active form of vitamin D, is dependent on 1α-hydroxylase in the proximal tubules of the kidney, which is regulated by the blood calcium concentration, parathyroid hormone (PTH), and fibroblast growth factor (FGF)-23 (Figure 3). In addition, 1,25(OH)_2_D acts as an active vitamin D, and 24-hydroxylase is activated in the proximal tubules of the kidney, inactivating 1,25(OH)_2_D to 1,24,25(OH)_3_D [68]. Similarly, 25(OH)D is inactivated by 24-hydroxylase to form 24,25(OH)_2_D [68] (Figure 3). Active vitamin D is strictly regulated by blood calcium concentration, PTH, FGF-23, and other factors, and blood 1,25(OH)_2_D and 25(OH)D concentrations are not necessarily correlated. 25(OH)D concentrations were measured when the vitamin D status was ascertained [68]. This is because the blood 25(OH)D concentration is approximately 1000 times easier to measure than the 1,25(OH)_2_D concentration, and its half-life is as long as 10 days to 3 weeks, which is thought to reflect the actual vitamin D status [71,72].

Recently, it has been well documented that hypovitaminosis D is often observed in dogs with CE [50,73,74,75,76,77,78,79]. In particular, hypovitaminosis D is more severe in dogs with PLE compared to CE dogs (normoalbuminemia) and is a poor prognostic factor [50,75]. Moreover, hypocalcemic seizures associated with hypovitaminosis D have been reported in PLE dogs [44,79]. Caution should be taken when using homemade ULFD for the treatment of dogs with PLE as there have also been reports of pathological fractures associated with secondary hyperparathyroidism due to further reduction in nutritional calcium intake [42]. However, there are few studies on highly evidence-based veterinary medicine.

To date, it is not known whether reduced vitamin D concentrations are the cause or only a consequence of intestinal diseases, such as dogs with CE and PLE [68]. A decrease in vitamin D intake or absorption due to the disease may be involved [68]. Indeed, inflammation of the gastrointestinal mucosa inhibits vitamin D absorption, resulting in lower serum vitamin D concentrations [68]. However, there is evidence that low concentrations of 25(OH)D may influence the development of an inflammatory bowel process, and 25(OH)D concentrations are negatively correlated with inflammatory mediators such as IL-8 in dogs with CE [68,77]. Therefore, vitamin D intake, absorption, and intestinal inflammation are closely associated.

## 2. Discussion

Recently, in the case of IRE, long-term follow-up showed little need for immunosuppressive doses of PSL or additional immunosuppressive drugs. In total, 73% (44/60) of IRE cases reported a change in diagnosis to FRE after one year [80]. Moreover, even in the case of steroid-resistant PLE, 80% (8/10) of dogs showed complete remission after switching to LFD [81]. These results suggest that dietary therapy is the most important strategy for treating dogs with CE and PLE.

Digestive enzymes and epithelial barrier functions are important for normal digestion [5,82]. However, it is difficult to assess their function, clinical severity, or prognosis. It has also been reported that intestinal permeability and mucosal absorptive capacity tests are not useful indicators for estimating the clinical disease activity in dogs with CE [83]. Therefore, in this manuscript, I focused on nutrition, particularly vitamins B_12_ and D, which are poor prognostic factors in dogs with CE and PLE.

Serum cobalamin concentrations were negatively correlated with ileal endoscopic and histological severity in dogs with CE [84]. However, the ideal cobalamin receptor is upregulated in dogs with CE [85]. This may explain why cobalamin works orally, even in dogs with CE and severe ileal inflammation. Therefore, cobalamin supplementation is a necessary and reasonable therapeutic strategy, even when administered orally. Recently, oral administrations of cobalamin supplementations have been reported to be effective not only for dogs with CE but also for dogs with PLE [86]. In contrast, serum MMA concentrations were increased in dogs with CE with hypocobalaminemia and have been reported to decrease with cobalamin supplementation [59,87]. However, increased serum MMA concentrations are sometimes observed even in dogs with normocobalaminemia, and one of the reasons is thought to be involved in intestinal dysbiosis [88]. One report described intestinal dysbiosis in dogs with CE with hypocobalaminemia compared to those with normocobalaminemia and healthy controls [89]. However, intestinal dysbiosis in follow-up samples remained in CE dogs that initially presented with hypocobalaminemia, even though they increased to normocobalamiinemia after 3 months of treatment with cobalamin supplementation [89]. These results suggest that intestinal dysbiosis may be associated with increased serum MMA concentrations without hypocobalaminemia. Similarly, it has been reported that there is no change in the dysbiosis index before and after standard treatment with dietary and immunosuppressive therapies in dogs with CE and PLE [90]. Thus, hypocobalaminemia, increased MMA concentrations, and intestinal dysbiosis are thought to be a complex combination in dogs with CE. Therefore, fecal microbiota transplantation (FMT) [91] may be a therapeutic tool for modulating intestinal dysbiosis and cobalamin metabolism in the future.

ULFD is deficient in vitamin D, which may cause nutritional problems with long-term use [44]. Therefore, a completely nutritional diet is the most important aspect of dietary therapy. The European Pet Food Industry Federation (FEDIAF) has issued guidelines for complete nutritional diets [https://europeanpetfood.org/wp-content/uploads/2024/09/FEDIAF-Nutritional-Guidelines_2024.pdf (accessed on 28 December 2024)]. The components of a complete nutritional diet include proteins (amino acids: arginine, histidine, isoleucine, leucine, lysine, methionine, cysteine, phenylalanine, tyrosine threonine, tryptophan, valine, and taurine), fat (linoleic acid, arachidonic acid, alpha-linolenic acid, eicosapentaenoic acid, and docosahexaenoic acid), minerals (calcium, phosphorus, potassium, sodium, chloride, and magnesium), trace elements (copper, iodine, iron, manganese, selenium, and zinc), and vitamins (vitamin A, vitamin D, vitamin E, vitamin B_1_, vitamin B_2_, vitamin B_5_, vitamin B_6_, vitamin B_12_, vitamin B_3_, vitamin B_9_, vitamin B_7_, choline, and vitamin K).

In a report that measured serum 25(OH)D concentration in correlation with intact PTH concentration by chemiluminescence immunoassay in healthy dogs, the optimal concentration was reported to be 100–120 ng/mL [92]; however, in dogs and cats, the reference value of 25(OH)D concentration in the blood has not been clearly established [68], and there are variations depending on the measurement method [78,79,93]. Therefore, in veterinary medicine, it is difficult to compare these values, and the development of a commercially established measurement method is desired. The vitamin D requirements of normal dogs and cats are also unknown, and according to the FEDIAF guidelines, 138–159 IU of cholecalciferol are required for 1000 kilocalories of metabolizable energy in normal dogs and 62.5–83.3 IU in normal cats. However, whether this intake provides sufficient 25(OH)D concentration has not yet been examined [68].

One case report of dogs with PLE and hypocalcemia showed that the serum 25(OH)D concentrations did not increase even though serum calcium concentrations improved and parathyroid hormone decreased after treatment with alfacalcidol, one of the active D3 vitamins that the carbon 1 has already hydroxylated [44]. It is thought that the drug exhibits physiological activity without the intervention of 25(OH)D; therefore, it might be considered that 25(OH)D concentrations did not improve even after vitamin D supplementation.

In humans, the anti-inflammatory role of vitamin D via the vitamin D receptor (VDR) has been well investigated in several diseases, including IBD [94]. Vitamin D inhibits the production of inflammatory cytokines via helper T cells in the gastrointestinal tract, suppresses the growth of harmful intestinal bacteria, and intervenes in intestinal mucosal cells by stimulating proteins involved in membrane junction integrity and intracellular pathogen recognition in human IBD [94]. Moreover, intestinal VDR expression is significantly decreased in patients with human IBD [95]. However, it has been reported that there was no statistical difference in duodenal VDR expression between healthy dogs and dogs with CE, in contrast to the findings in humans [96]. This may explain why vitamin D works orally, even in dogs with PLE and intestinal inflammation. Moreover, vitamin D supplementation may elicit an anti-inflammatory response through its interaction with VDR. Therefore, oral vitamin D supplementation may be a potential therapeutic strategy for dogs with CE. Although vitamin D supplementation has not yet been determined as a clear protocol for dosage or whether it improves prognosis, it is a reasonable treatment for dogs with PLE with hypocalcemia.

## 3. Conclusions

Canine CE has multiple and different etiologies. Cobalamin (vitamin B_12_) and vitamin D are poor prognostic factors in dogs with CE and PLE; therefore, cobalamin supplementation in dogs with CE and vitamin D supplementation in PLE dogs with hypocalcemia may be a reasonable therapeutic strategy. The main treatment strategy for dogs with CE and PLE is complete nutritional dietary therapy.

## 4. Future Directions

Clinical trials are needed to determine whether these vitamin supplements are useful for improving the prognosis of dogs with CE and PLE.

## Figures and Tables

**Figure 1 animals-15-00649-f001:**
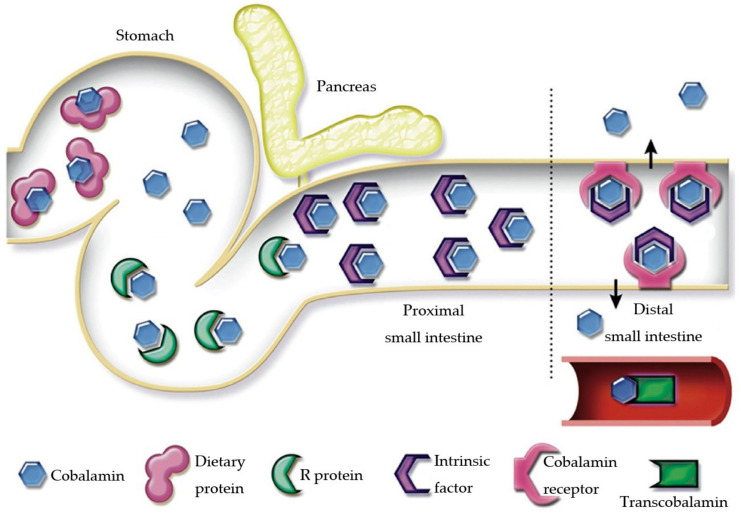
Absorption of cobalamin by carnivores. Normal absorption of cobalamin requires the function of the stomach (acid, proteases, and IF), pancreatic exocrine (IF and proteases), and ileal mucosa (receptors for cobalamin plus IF). IF, intrinsic factors. Adapted from Ruaux, 2013 [54].

**Figure 2 animals-15-00649-f002:**
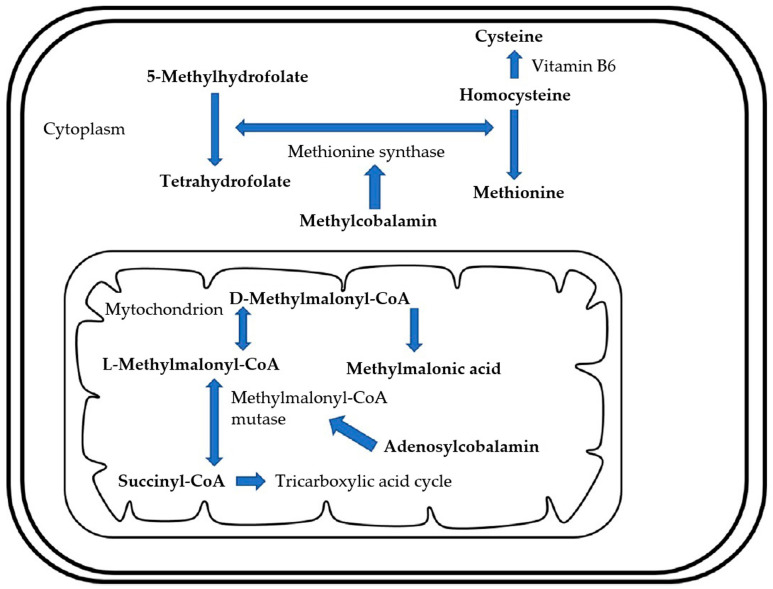
The major intracellular metabolic pathway requires cobalamin. In the mitochondria, adenosylcobalamin is required as a cofactor for methylmalonyl-CoA mutase, which converts L-methylmalonyl-CoA to succinyl-CoA. In the cytoplasm, methylcobalamin is required as a cofactor for methionine synthase, which converts homocysteine to methionine and 5-methylhydrofolate to tetrahydrofolate. Adapted from Toresson et al., 2019 [59].

**Figure 3 animals-15-00649-f003:**
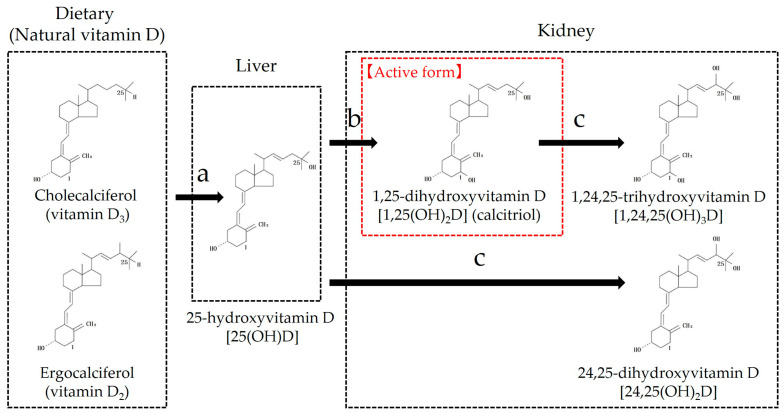
Vitamin D metabolism. (**a**) Natural vitamin D is transported to the liver by VDBP and is converted to 25-hydroxyvitamin D [25(OH)D] by 25-hydroxylase. The biological activity of 25(OH)D_3_ derived from vitamin D_3_ (cholecalciferol) was better than that of 25(OH)D_2_ derived from vitamin D_2_ (ergocalciferol) because of its higher affinity for VDBP. This reaction is not strictly regulated and is thought to proceed in a substrate-dependent manner. (**b**) 25(OH)D is converted to 1,25-dihydroxyvitamin D [1,25(OH)_2_D], an active form of vitamin D, by 1α-hydroxylase in the proximal tubule of the kidney. This reaction is tightly regulated, stimulated by PTH and hypophosphatemia, and inhibited by FGF-23 and 1,25(OH)_2_D, hypercalcemia, and hyperphosphatemia. (**c**) 25(OH)D and 1,25(OH)_2_D are inactivated by 24-hydroxylase in the proximal tubules of the kidney. FGF, fibroblast growth factor; PTH, parathyroid hormone; VDBP, vitamin D-binding protein.

**Table 1 animals-15-00649-t001:** CIBDAI and CCECAI score. The above six findings (attitude/activity, appetite, vomiting, stool consistency, stool frequency, and weight loss) are CIBDAI scores.

Findings\Score	0	1	2	3
Attitude/activity	normal	slightly decreased	moderately decreased	severely decreased
Appetite	normal	slightly decreased	moderately decreased	severely decreased
Vomiting	normal	mild (1×/wk)	moderate (2–3×/wk)	severe (>3×/wk)
Stool consistency	normal	slightly soft feces	very soft feces	watery diarrhea
Stool frequency	normal	slightly increased (2–3×/d) or fecal blood, mucus, or both	moderately increased (4–5×/d)	severely increased (>5×/d)
Weight loss	none	mild (<5%)	moderate (5–10%)	severe (>10%)
Albumin levels	>2.0 g/dL	1.5–1.99 g/dL	1.2–1.49 g/dL	<1.2 g/dL
Ascites and peripheral edema	none	mild ascites or peripheral edema	moderate ascites/ peripheral edema	severe ascites/ pleural effusion and peripheral edema
Pruritus	no pruritus	occasional episodes of itching	regular episodes of itching, but stops when dog is asleep	dog regularly wakes up because of itching

Severity was defined as below: insignificant disease, 0–3; mild disease, 4–5; moderate disease, 6–8; severe disease, 9–11; very severe disease, >12. CIBDAI, canine inflammatory bowel disease activity index; CCECAI, canine chronic enteropathy activity index. Adapted from Jergens et al., 2003 [10] and Allenspach et al., 2007 [6].

**Table 3 animals-15-00649-t003:** Cobalamin amounts in dogs.

	Body Weight (kg)							Route	Reference
	<5	5–10	10–20	20–30	30–40	40–50	>50		
Cobalamin amounts (µg)	250	400	600	800	1000	1200	1500	SC	Ruaux, 2013 [54]
	250		500	1000				PO	Toresson et al., 2016 [63] Toresson et al., 2019 [59]

PO, per os; SC, subcutaneous.

## Data Availability

No data were used for the research described in this article.

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
