# Peer review of "Chronic Enteropathy and Vitamins in Dogs"

_animals, 2025, doi:10.3390/ani15050649_

Round 1

Reviewer 1 Report

Comments and Suggestions for Authors

Overall comments

Thank for you the opportunity to review this interesting and relevant review paper exploring the literature and findings relating to canine chronic enteropathy and possible dietary support.

This review is very readable and comprehensive paper, covering some key studies and combining them into a useful and accessible resource that I suspect will be useful for veterinary surgeons, paraprofessionals, and caregivers. There is a tendency to be descriptive rather than critical with regards to the content, but I do not feel this is a problem or detracts from the work and its value – rather it reflects the nature of this review paper.

Overall, this is a well written and considered review with only a few areas of inconsistency in approach and content delivery – I found it logical and with good clarity and narrative. Tables and diagrams are suitable and appropriate.

The background consideration is suitable to set the scene and identify intent of the work. Conclusions and discussion are pertinent to the review.

I highlight some points that are noted in comments on the PDF also;

1.    Please consider the use of the term ‘dose’ – I appreciate this might be used in the original literature BUT it can be a problematic term in feed supplementation as it can be construed as a pharmaceutical term (and this is not appropriate) – can lead to commercial and caregiver confusion that supplementation is a treatment – leads to many regulatory problems (certainly in the UK).

2.    Be careful with specificity about wording to ensure full clarity.

3.    Nutritional guidelines not mentioned and the importance of any diet for treatment/management/therapeutic support to be nutritionally complete might be worth noting.

References

I have not exhaustively gone through these, but all appear fine although I have not proofread or cross referenced to check validity of use.

Author Response

Dear reviewer,

Thank you for your kind and valuable comments.

I corrected my manuscript according to your advice.

The revised part is being shown in red colors.

  1. Please consider the use of the term ‘dose’ – I appreciate this might be used in the original literature BUT it can be a problematic term in feed supplementation as it can be construed as a pharmaceutical term (and this is not appropriate) – can lead to commercial and caregiver confusion that supplementation is a treatment – leads to many regulatory problems (certainly in the UK).

Thank you for your advice. Based on your valuable suggestions, I corrected the term ‘dose’ to ‘amount’.

  1. Be careful with specificity about wording to ensure full clarity.

Thank you for your kindness. I have replied to the PDF file.

  1. Nutritional guidelines not mentioned and the importance of any diet for treatment/management/therapeutic support to be nutritionally complete might be worth noting.

Thank you for your valuable advice. I described the importance of perfect nutritional diets. Then, I added the sentences, as below.

Line 325-336: ULFD is not a complete nutritional diet, and its long-term use can cause nutritional problems [45]. Therefore, a complete nutritional diet is the most important aspect of dietary therapy. European Pet Food Industry Federation (FEDIAF) has issued guidelines for complete nutritional diets (https://europeanpetfood.org/wp-content/uploads/2024/09/FEDIAF-Nutritional-Guidelines_2024.pdf). The components of a complete nutritional diet include protein (amino acids; arginine, histidine, isoleucine, leucine, lysine, methionine, cysteine, phenylalanine, tyrosine threonine, tryptophan, valine, taurine), fat (linoleic acid, arachidonic acid, alpha-linolenic acid, eicosapentaenoic acid, docosahexaenoic acid), minerals (calcium, phosphorus, potassium, sodium, chloride, magnesium), trace elements (copper, iodine, iron, manganese, selenium, zinc), and vitamins (vitamin A, vitamin D, vitamin E, vitamin B1, vitamin B2, vitamin B5, vitamin B6, vitamin B12, vitamin B3, vitamin B9, vitamin B7, choline, vitamin K).

Reviewer 2 Report

Comments and Suggestions for Authors

Dear author,

it is positive that this manuscript “Chronic Enteropathy and Vitamins in Dogs” (animals-3424702) by Yu Tamura devoted to the important and modern topics. The author postulated that “…cobalamin (vitamin B12) and vitamin D" is considered "negative prognostic factors for chronic enteropathy in dogs". That is why, the author represented these vitamins as the key elements of the "treatment strategy about Chronic Enteropathy (CE)”. It is well-known that “dietary therapy is the first step in the treatment of chronic enteropathy”. That is why, some particular vitamins “should be supplemented” in the diet for animals. On the other hand, the author described this disease in detail and divided it into the following classes: “Food-responsive enteropathy (FRE)”, “Antibiotic-responsive enteropathy (ARE)”, “Immunosuppressant-responsive enteropathy (IRE)”, etc. and discussed their therapy. The author presented numerous data concerning the vitamin B12 and vitamin D, which have, of course, particular importance for “home dietary therapy” for the dogs and other animals.

The manuscript analyze the literature works in detail and at high level of discussion. I do not doubt the technical quality of the work and feel that there is a sufficient impact on a broader readership to justify publication in the "Animals". This topic is in frame of the journal scopes, the subject matter is treated in depth.

There are some comments:

1. It is difficult to follow the major ideas and trends in this manuscript, because there are only three mandatory parts (“Introduction”, “Discussion”, “Conclusions”) in this manuscript without subsections (with subtitles). It will be reasonable to make some structuring (precise subsections) in the part 1 “Introduction” and in the part 2. “Discussion”.  In particular, the subsection 1.1. with subtitle “Food-responsive enteropathy (FRE) and its therapy” (on the page 2, from line 60); the subsection 1.2. with subtitle “Antibiotic-responsive enteropathy (ARE) and its therapy” (on the page 3); the subsection 1.3. with subtitle “Immunosuppressant-responsive enteropathy (IRE) and its therapy” (on the page 4); etc.

2. Please, change, at the bottom line of the scheme of mitochondria (see Fig.2, on the page 7), the words “tricarboxylic acid cycle” (TCA) or “citric acid cycle” instead of the wrong word “tricarbocyclic” acid cycle”.

3. It will be reasonable to convert the Table 3 (“Cobalamin dosage in dogs”) into the plain text, because this Table 3 is too short (small in lines) and is not understandable without additional explanations in text.

4. Please, check the chemical formulae at the scheme on the Fig.3 (page 8). It concerns the formulae of the 25-hydroxyvitamin D3, 1,25-dihydroxyvitamin D3, etc., because all of them on the Fig.3 have chemical formula of the vitamin D2 (as derivatives of the ergocalciferol), but must be derivatives of the cholecalciferol (vitamin D3).

5. Concerning the part 2. “Discussion”:

5.1. The authors discuss chronic diarrhea, but do not describe or discuss in details the mechanisms of the digestion and absorption processes. There is almost no assessment of the coordination of the action of digestive enzymes and nutrient transporters. The barrier function of the intestine depends on the tightness of the epithelial cells, which in turn depends on the organization of tight junctions. Tight junctions, located on the apical side of enterocytes, are multiprotein complexes consisting of transmembrane proteins such as occludin, claudins (claudin-1, claudin 2, claudin 3), tricellulin, and intercellular adhesion molecules associated with cytosolic molecules such as zona occludens proteins (ZO-1, ZO-2, and ZO-3). The amount of these molecules is directly related to the changes in intestinal permeability. There is almost nothing about this in the paper.

5.2. The characteristics for monitoring the epithelial barrier and digestion are: villus height, gene expression and/or tight junction protein levels, goblet cell or mucin counts, digestive enzyme activity, nutrient transporters, cell proliferation, diarrhea occurrence, intestinal permeability, cell apoptosis. Why (for what reasons) did the authors ignore the above and focus only on vitamin B12 and D ?

5.3. Why did the author not consider the role of protein nutrition and individual amino acids in this work? Since amino acids play a major role in the structural, energetic, immune and antioxidant functions for the intestine and it will be reasonable to discuss such topics in the text.

Thus, the present manuscript can be accepted after minor revisions.

Author Response

Dear reviewer,

Thank you for your kind and valuable comments.

I corrected my manuscript according to your advice.

The revised part is being shown in red colors.

  1. It is difficult to follow the major ideas and trends in this manuscript, because there are only three mandatory parts (“Introduction”, “Discussion”, “Conclusions”) in this manuscript without subsections (with subtitles). It will be reasonable to make some structuring (precise subsections) in the part 1 “Introduction” and in the part 2. “Discussion”. In particular, the subsection 1.1. with subtitle “Food-responsive enteropathy (FRE) and its therapy” (on the page 2, from line 60); the subsection 1.2. with subtitle “Antibiotic-responsive enteropathy (ARE) and its therapy” (on the page 3); the subsection 1.3. with subtitle “Immunosuppressant-responsive enteropathy (IRE) and its therapy” (on the page 4); etc.

Thank you for your advice. Based on your valuable suggestions, I corrected the manuscript as bellow.

Line 55: 1.1. Food-responsive enteropathy (FRE) and its therapy

Line 96: 1.2. Antibiotic-responsive enteropathy (ARE) and its therapy

Line 113: 1.3. Immunosuppressant-responsive enteropathy (IRE) and its therapy

Line 160: 1.4. Non-responsive enteropathy (NRE) and its therapy

Line 177: 1.5. Protein-losing enteropathy (PLE) and its therapy

Line 197: 1.6. Prognostic factors in dogs with CE and PLE

Line 219: 1.7. Vitamin B12 in dogs with CE

Line 264: 1.8. Vitamin D in dogs with PLE

  1. Please, change, at the bottom line of the scheme of mitochondria (see Fig.2, on the page 7), the words “tricarboxylic acid cycle” (TCA) or “citric acid cycle” instead of the wrong word “tricarbocyclic” acid cycle”.

Thank you for your advice. I corrected it.

  1. It will be reasonable to convert the Table 3 (“Cobalamin dosage in dogs”) into the plain text, because this Table 3 is too short (small in lines) and is not understandable without additional explanations in text.

Thank you for your advice. I revised and added the sentences as below.

Line 246-250: The SC supplemented amounts are reported 250-1200 µg of hydroxocobalamin depending on body weight [55], and the PO feeding amounts are reported in dogs with body weight of 1–10 kg received 250 µg, 10–20 kg received 500 µg, and >20 kg received 1000 µg daily [63, 64] (Table 3).

  1. Please, check the chemical formulae scheme on the Fig.3 (page 8). It concerns the formulae of the 25-hydroxyvitamin D3, 1,25-dihydroxyvitamin D3, etc., because all of them on the Fig.3 have chemical formula of the vitamin D2 (as derivatives of the ergocalciferol), but must be derivatives of the cholecalciferol (vitamin D3).

Thank you for your critical advice. As you indicated, the chemical formulae scheme was misleading and has been corrected. Moreover, I added the sentence, as below.

Line 302-304: The biological activity of 25(OH)D3 derived from vitamin D3 (cholecalciferol) is better than that of 25(OH)D2 derived from vitamin D2 (ergocalciferol) because of its higher affinity with VDBP.

  1. Concerning the part 2. “Discussion”:

5.1. The authors discuss chronic diarrhea, but do not describe or discuss in details the mechanisms of the digestion and absorption processes. There is almost no assessment of the coordination of the action of digestive enzymes and nutrient transporters. The barrier function of the intestine depends on the tightness of the epithelial cells, which in turn depends on the organization of tight junctions. Tight junctions, located on the apical side of enterocytes, are multiprotein complexes consisting of transmembrane proteins such as occludin, claudins (claudin-1, claudin 2, claudin 3), tricellulin, and intercellular adhesion molecules associated with cytosolic molecules such as zona occludens proteins (ZO-1, ZO-2, and ZO-3). The amount of these molecules is directly related to the changes in intestinal permeability. There is almost nothing about this in the paper.

5.2. The characteristics for monitoring the epithelial barrier and digestion are: villus height, gene expression and/or tight junction protein levels, goblet cell or mucin counts, digestive enzyme activity, nutrient transporters, cell proliferation, diarrhea occurrence, intestinal permeability, cell apoptosis. Why (for what reasons) did the authors ignore the above and focus only on vitamin B12 and D ?

Thank you for your valuable advice. I think the digestion and absorption processes, digestive enzymes and nutritional transporters, and the barrier function are certainly important mechanisms. However, from the perspective of poor prognosis, which is clinically important, I have focused my discussion on vitamin B12 and vitamin D. Therefore, I added the sentences, as below.

Line 319-324: Digestive enzymes and epithelial barrier function are important for normal digestion [77, 78]. However, it is difficult to assess their function and clinical severity and prognosis. There is also a report that intestinal permeability and mucosal absorptive capacity tests were not useful indicators for estimating clinical disease activity in dogs with CE [79]. Therefore, I focused on nutrition in this manuscript, particularly vitamin B12 and vitamin D, which are poor prognostic factors in dogs with CE and PLE.

5.3. Why did the author not consider the role of protein nutrition and individual amino acids in this work? Since amino acids play a major role in the structural, energetic, immune and antioxidant functions for the intestine and it will be reasonable to discuss such topics in the text.

Thank you for your valuable advice. I think protein nutrition is also important. Therefore, I described the importance of perfect nutritional diets. Then, I added the sentences, as below.

Line 325-336: ULFD is not a complete nutritional diet, and its long-term use can cause nutritional problems [45]. Therefore, a complete nutritional diet is the most important aspect of dietary therapy. European Pet Food Industry Federation (FEDIAF) has issued guidelines for complete nutritional diets (https://europeanpetfood.org/wp-content/uploads/2024/09/FEDIAF-Nutritional-Guidelines_2024.pdf). The components of a complete nutritional diet include protein (amino acids; arginine, histidine, isoleucine, leucine, lysine, methionine, cysteine, phenylalanine, tyrosine threonine, tryptophan, valine, taurine), fat (linoleic acid, arachidonic acid, alpha-linolenic acid, eicosapentaenoic acid, docosahexaenoic acid), minerals (calcium, phosphorus, potassium, sodium, chloride, magnesium), trace elements (copper, iodine, iron, manganese, selenium, zinc), and vitamins (vitamin A, vitamin D, vitamin E, vitamin B1, vitamin B2, vitamin B5, vitamin B6, vitamin B12, vitamin B3, vitamin B9, vitamin B7, choline, vitamin K).

Reviewer 3 Report

Comments and Suggestions for Authors

Review report of the manustript “Chronic Enteropathy and Vitamins in Dogs” by Yu Tamura

General comments and final considerations:

Personally, I find this manuscript not suitable for the publication as for how it is presented at the moment.
There are some major issues:
- The construction of both introduction and discussion and chaotic and not focused on the main aim of the review (the role of vitamins D and B12 in canine CE).
- There are many referral to old publications, classifications and therapeutic strategies (I would recommend to refer to more recent literature).
- Many assertive assumptions are made from studies with a low “evidence based medicine” grade, I would suggest to be more moderate, in such cases.
- By reading the whole manuscript, the reader is still confused about what is the current literature regarding the vitaminic profile of these kind of patients, which can be the pathophysiological mechanisms involved, what is the clinical evidence of the potential benefits.
- There are many English mistakes and the construction of the sentences sometimes make them quite difficult to understand.
- The manuscript body lacks of connections between the various concepts and it gives confusing information which are not strictly necessary, resulting in a paper which is actually difficult to follow and with a “broken” flow.

I would suggest to the author to deeply revise and re-structure the manuscript and eventually resubmit it after those modifications, which could for sure ameliorate this review. The aim of this review is to overview a topic which is actually very interesting and of clinical relevance and importance.

Following, some specific comments:

Simple Summary

Lines 8-9: “Acute gastroenteritis improves quickly with symptomatic treatment...”
I personally believe that this statement is not accurate, considering acute enteropathies, I would not state in that assertive way that they improve with symptomatic treatment; it may be true, however, considering the aetiology (i.e, parvoviral) or the potential risk of SIRS-sepsis development in these cases (due to alteration of intestinal wall permeability and leaky-gut syndrome), I would suggest to modify this statement and be more moderate.

Lines 12-13: “I hope this will be also helpful for home dietary therapy for the owners
I don’t understand the meaning of this sentence; it may be unclear. Moreover, I would be careful to refer to the owners since from this sentence people can assume that the vitaminic supplementation should or could be modified by the owners independently from the veterinary recommendation. I would rather suggest referring to the potential useful insights that this review may provide to clinicians in the nutritional management of these patients, eventually both those treated with home-cooked or commercial diets.

Abstract

Line 16: I will suggest to replace “and so on” with “persisting”

Line 18-20: “The diagnosis is determined by excluding other diseases such as parasitic infections, tumors, pancreatitis, pancreatic exocrine insufficiency, metabolic diseases, endocrine diseases such as hypoadrenocorticism, and so on”
I agree with this statement, however if you refer to chronic inflammatory enteropathies (as for I can assume) I would also refer to the need of an histological confirmation (if you refer to chronic inflammatory enteropathies). i.e. “The diagnosis is histologically confirmed, after excluding other diseases…”
(Dupouy-Manescau N, Méric T, Sénécat O, Drut A, Valentin S, Leal RO, Hernandez J. Updating the Classification of Chronic Inflammatory Enteropathies in Dogs. Animals (Basel). 2024 Feb 21;14(5):681. doi: 10.3390/ani14050681. PMID: 38473066; PMCID: PMC10931249.)

Line 20-25: I would also recommend mentioning those who were recently identified as microbiota-modulation responsive enteropathies (i.e. those responsive to pro/pre/synbiotic products or fecal microbial transplantation)
(Dupouy-Manescau N, Méric T, Sénécat O, Drut A, Valentin S, Leal RO, Hernandez J. Updating the Classification of Chronic Inflammatory Enteropathies in Dogs. Animals (Basel). 2024 Feb 21;14(5):681. doi: 10.3390/ani14050681. PMID: 38473066; PMCID: PMC10931249.)

Line 28: “Therefore, this review represents treatment strategy about CE and these vitamins.”
This sentence is unclear and personally, too assertive. I would eventually say that this review may provide a current state of literature regarding the alterations and therapeutic potential in CE dogs.

General comment: the abstract is same as the first part of the introduction; I would suggest to deeply re-arrange the abstract, maybe providing information about 1. CE definition 2. Focus on the complexity of its nutritional management (considering malabsorption, dysbiosis, inflammation, …) 3. Current state of literature and guidelines regarding vitamin supplementation 4. Aim of the review

Introduction
Line 32: I would not say “unknown” cause multiple and different aetiologies and causes are identified. I would remove this term since I personally believe it not to be accurate.

Line 35: I would state that the diagnosis “further requires to rule out other diseases…”

Lines 38-47: In this section I would recommend to refer also to the microbiota-modulation responsive enteropathies (i.e. those responsive to pro/pre/synbiotic products or fecal microbial transplantation). I will suggest to rephrase this paragraph according to the latest classification of chronic inflammatory enteropathies.
(Dupouy-Manescau N, Méric T, Sénécat O, Drut A, Valentin S, Leal RO, Hernandez J. Updating the Classification of Chronic Inflammatory Enteropathies in Dogs. Animals (Basel). 2024 Feb 21;14(5):681. doi: 10.3390/ani14050681. PMID: 38473066; PMCID: PMC10931249)

Lines 49-56: I personally think that this part is unnecessary; you already mentioned before the importance of excluding parasitic infection and I think that further discuss those aspects may be redundant considering the aim of the review.
Moreover, in a review, it may be inappropriate to cite a personal “case report” considering that reviews should be considered as one of the highest sources of accurate scientific information (in terms of evidence-based medicine). Thus, for a review, I will eventually recommend citing and refer to solid literature rather than personal experience and evidence, even if it still represents something very important and interesting.
I would suggest to directly remove this paragraph, if not strictly necessary.

Lines 56-59: Similarly, I would suggest not to go deeper in the diagnostic criteria of canine hypoadrenocorticism since it is not strictly pertinent to the review’ aim and you already mentioned the importance of excluding this endocrine disorder in the diagnostic process of chronic inflammatory enteropathies. I would suggest to remove this part if not strictly necessary, or, at least, cite the latest guidelines for the diagnosis.

Lines 60-86: This paragraph font is smaller than the remaining manuscript.

Lines 60-189: Moreover this paragraph is chaotic and should be rephrased. I would rather suggest to simplify it and provide brief but clear information regarding the diagnostic iter (based on treatment response) in their subsequential steps without citing too many exceptions (i.e Chronic idiopathic colon diarrhea). This review’s aim is not to overview “chronic enteropathies” (or “chronic inflammatory enteropathies”, please eventually check and modify accordingly to which entity you are referring in all the manuscript) but to focus on vitaminic status. Thus, I will reccomend to significantly reduce this introductive part and eventually focus on the mechanism which can cause vitaminic alteration and their potential therapeutic role according to the latest literature.

Line 190: In this point I would suggest to discuss the poor prognostic factors associated with chronic inflammatory enteropathies, before going in the particular of PLE’s ones.

Line 226: After this part “the most important factor [54]” I would suggest to place a brief explanation about the cobalamin status in canine CE. You directly discuss about the supplementation right after the explanation of the metabolism of cobalamin, but a fundamental junction element is the cobalamin status of these patients.

Line 238: “sufficient” is an aspecific term, I would suggest to refer to the function without specifying.

Lines 242: The font used in Figure 1 and 2 is different, please correct it according to the template guidelines.

Lines 261: “Due to different diet” is not maybe correct, cause this not a cause rather than the consequence of different Vitamin D metabolism.

Lines 274-278: This part is unclear, I would suggest rephrasing to make it easier to understand.

Lines 279-284: You state that the hypovitaminosis D in PLE dogs is still in research phase, but I would suggest to consider that this state is well documented in CE dogs; another time, before discussing about its potential prognostic factor, I would rather recommend to talk about the general vitamin D status in dogs with CE (not only PLE), and after, about its potential prognostic role.

Discussion
Lines 296-301: Sorry, I personally can’t understand the role of this paragraph, placed here. It is still discussing about specific therapeutic aspects, not mentioning the vitaminic profile and role.

Lines 302-308: It is not easy to follow this part and the relationship between intestinal dysbiosis and cobalamin should be more extensively explained, it could be an interesting aspect to discuss.

Line 308: “Similarly, it has been reported that there is no change in dysbiosis index before and after treatment in CE dogs with PLE”, which treatment?

Line 309-310: “Therefore, fecal microbiota transplantation (FMT) [78] may be a therapeutic strategy for a radical solution to intestinal dysbiosis in the future.”
How this assumption is related to the rest of the paragraph? It seems disconnected and decontextualized. Moreover, it is a strong assumption, I would rather suggest to be more positive and stating that FMT can eventually be considered as a therapeutical tool to modulate the intestinal microbiota (“radical solution” is a strong assumption).

Lines 314-345: “dogs and cats” is repeated twice in the same sentence.

Lines 321-322: “However, it has not been examined whether this intake is sufficient for 25(OH)D levels in blood”; sufficient to reach what level? This sentence’s meaning is confusing, please rephrase it.

Lines 329-331: “Although vitamin D supplementation is not yet clear whether a clear protocol for dosage can be established or whether it improves prognosis, it is certainly a reasonable treatment for PLE dogs with hypocalcemia
This sentence has issues in its construction which makes it difficult to understand. Moreover, I would remove the word “certainly” since it underscore a certain grade of scientific evidence which you state as not sufficient in the sentence just before this assumption.

Conclusions
Line 333: I would not use the term unknown (see comment above).

Line 334-335: “Cobalamin and vitamin D are poor prognostic factors in dogs with CE and PLE; therefore, cobalamin supplement in dogs with CE and vitamin D supplement in PLE dogs with hypocalcemia is important.”
I personally believe that this sentence may be conceptually wrong; if a parameter is a poor prognostic factor, it doesn’t necessarily mean that it’s supplementation would be beneficial. Clinical trials are needed to assess whether these supplementations may be useful or not.
I would eventually say that “clinical trials showed that vit.D and B12 supplementations provide benefits to CE dogs”, citing the related studies which can support this assumption. 

Future directions

Lines 338-339: I can’t understand how this assumption may be related to the aim and presumptive main focus of the review (vitamins).

Comments on the Quality of English Language

There are many English mistakes and the construction of the sentences sometimes make them quite difficult to understand.

Author Response

Dear reviewer,

Thank you for your kind and valuable comments.

I corrected my manuscript according to your valuable advice.

The revised part is being shown in red colors.

Personally, I find this manuscript not suitable for the publication as for how it is presented at the moment.

There are some major issues:

- The construction of both introduction and discussion and chaotic and not focused on the main aim of the review (the role of vitamins D and B12 in canine CE).

- There are many referral to old publications, classifications and therapeutic strategies (I would recommend to refer to more recent literature).

- Many assertive assumptions are made from studies with a low “evidence based medicine” grade, I would suggest to be more moderate, in such cases.

- By reading the whole manuscript, the reader is still confused about what is the current literature regarding the vitaminic profile of these kind of patients, which can be the pathophysiological mechanisms involved, what is the clinical evidence of the potential benefits.

- There are many English mistakes and the construction of the sentences sometimes make them quite difficult to understand.

- The manuscript body lacks of connections between the various concepts and it gives confusing information which are not strictly necessary, resulting in a paper which is actually difficult to follow and with a “broken” flow.

I would suggest to the author to deeply revise and re-structure the manuscript and eventually resubmit it after those modifications, which could for sure ameliorate this review. The aim of this review is to overview a topic which is actually very interesting and of clinical relevance and importance.

Thank you for your advice. Based on your valuable suggestions, I made some structuring (precise subsections) in introduction and revised with more recent literature throughout the paper. Moreover, I plan to submit the manuscript for professional English editing.

Lines 8-9: “Acute gastroenteritis improves quickly with symptomatic treatment...”

I personally believe that this statement is not accurate, considering acute enteropathies, I would not state in that assertive way that they improve with symptomatic treatment; it may be true, however, considering the aetiology (i.e, parvoviral) or the potential risk of SIRS-sepsis development in these cases (due to alteration of intestinal wall permeability and leaky-gut syndrome), I would suggest to modify this statement and be more moderate.

Thank you for your advice. I revised the sentence, as below.

Line 8-9: Transient acute gastroenteritis improves quickly with symptomatic treatment; however, if symptoms persist for more than 3 weeks, chronic enteropathy should be considered.

Lines 12-13: “I hope this will be also helpful for home dietary therapy for the owners”

I don’t understand the meaning of this sentence; it may be unclear. Moreover, I would be careful to refer to the owners since from this sentence people can assume that the vitaminic supplementation should or could be modified by the owners independently from the veterinary recommendation. I would rather suggest referring to the potential useful insights that this review may provide to clinicians in the nutritional management of these patients, eventually both those treated with home-cooked or commercial diets.

Line 16: I will suggest to replace “and so on” with “persisting”

Thank you for your advice. I deleted the sentence.

Line 18-20: “The diagnosis is determined by excluding other diseases such as parasitic infections, tumors, pancreatitis, pancreatic exocrine insufficiency, metabolic diseases, endocrine diseases such as hypoadrenocorticism, and so on”

I agree with this statement, however if you refer to chronic inflammatory enteropathies (as for I can assume) I would also refer to the need of an histological confirmation (if you refer to chronic inflammatory enteropathies). i.e. “The diagnosis is histologically confirmed, after excluding other diseases…”

(Dupouy-Manescau N, Méric T, Sénécat O, Drut A, Valentin S, Leal RO, Hernandez J. Updating the Classification of Chronic Inflammatory Enteropathies in Dogs. Animals (Basel). 2024 Feb 21;14(5):681. doi: 10.3390/ani14050681. PMID: 38473066; PMCID: PMC10931249.)

Line 20-25: I would also recommend mentioning those who were recently identified as microbiota-modulation responsive enteropathies (i.e. those responsive to pro/pre/synbiotic products or fecal microbial transplantation)

(Dupouy-Manescau N, Méric T, Sénécat O, Drut A, Valentin S, Leal RO, Hernandez J. Updating the Classification of Chronic Inflammatory Enteropathies in Dogs. Animals (Basel). 2024 Feb 21;14(5):681. doi: 10.3390/ani14050681. PMID: 38473066; PMCID: PMC10931249.)

Line 28: “Therefore, this review represents treatment strategy about CE and these vitamins.”

This sentence is unclear and personally, too assertive. I would eventually say that this review may provide a current state of literature regarding the alterations and therapeutic potential in CE dogs.

General comment: the abstract is same as the first part of the introduction; I would suggest to deeply re-arrange the abstract, maybe providing information about 1. CE definition 2. Focus on the complexity of its nutritional management (considering malabsorption, dysbiosis, inflammation, …) 3. Current state of literature and guidelines regarding vitamin supplementation 4. Aim of the review

Thank you for your valuable advice. According to your great advice, I revised the abstract, as below.

Line 14-27: Chronic enteropathy (CE) or chronic inflammatory enteropathy is a group of diseases which have multiple and different etiologies characterized by chronic gastrointestinal signs such as vomiting, diarrhea, anorexia, and weight loss for more than 3 weeks, and inflammatory cell infiltration such as lymphoplasmacytic cells in the intestinal mucosal lamina propria. The diagnosis is histologically confirmed, after excluding other diseases such as parasitic infections, tumors, pancreatitis, exocrine pancreatic insufficiency, metabolic diseases, endocrine diseases such as hypoadrenocorticism. Nutritional management depends on several important functions such as digestion and absorption processes, digestive enzymes and nutritional transporters, and the barrier function. Moreover, intestinal dysbiosis may have been found to be involved in a variety of functions. Recently, cobalamin (vitamin B12) and vitamin D have been considered negative prognostic factors in dogs with CE. Cobalamin supplementation ameliorates clinical disease severity in dogs with CE and vitamin D supplementation ameliorates hypocalcemic status in CE dogs with hypoalbuminemia. Therefore, the aim of this review is overview of CE, and represents treatment and nutritional management strategy about CE and prognostic vitamins.

Line 32: I would not say “unknown” cause multiple and different aetiologies and causes are identified. I would remove this term since I personally believe it not to be accurate.

Line 35: I would state that the diagnosis “further requires to rule out other diseases…”

Thank you for your advice. I revised the sentences, as below.

Lines 31-37: Chronic enteropathy (CE) or chronic inflammatory enteropathy is a group of diseases which have multiple and different etiologies characterized by chronic gastrointestinal signs such as vomiting, diarrhea, anorexia, and weight loss for more than 3 weeks and inflammatory cell infiltration such as lymphoplasmacytic cells in the intestinal mucosal lamina propria [1-3]. The diagnosis further requires to rule out other diseases such as parasitic infections, tumors, pancreatitis, exocrine pancreatic insufficiency, metabolic diseases, endocrine diseases such as hypoadrenocorticism.

Lines 38-47: In this section I would recommend to refer also to the microbiota-modulation responsive enteropathies (i.e. those responsive to pro/pre/synbiotic products or fecal microbial transplantation). I will suggest to rephrase this paragraph according to the latest classification of chronic inflammatory enteropathies.

(Dupouy-Manescau N, Méric T, Sénécat O, Drut A, Valentin S, Leal RO, Hernandez J. Updating the Classification of Chronic Inflammatory Enteropathies in Dogs. Animals (Basel). 2024 Feb 21;14(5):681. doi: 10.3390/ani14050681. PMID: 38473066; PMCID: PMC10931249)

Thank you for your valuable advice. I revised and added the sentences, as below.

Line 47-48: Moreover, ARE is proposed to be replacing with microbiota-related modulation-responsive enteropathy (MrMRE) [2].

Line 110-113: Recently, ARE was proposed to be replaced with microbiota-related modula-tion-responsive enteropathy (MrMRE) [2]. This category would be proposed including enteropathy responding to pre/pro/symbiotic products or fecal microbiota transplantation although there is not yet reliable diagnostic criterion for MrMRE category [2].

Lines 49-56: I personally think that this part is unnecessary; you already mentioned before the importance of excluding parasitic infection and I think that further discuss those aspects may be redundant considering the aim of the review.

Moreover, in a review, it may be inappropriate to cite a personal “case report” considering that reviews should be considered as one of the highest sources of accurate scientific information (in terms of evidence-based medicine). Thus, for a review, I will eventually recommend citing and refer to solid literature rather than personal experience and evidence, even if it still represents something very important and interesting.

I would suggest to directly remove this paragraph, if not strictly necessary.

Lines 56-59: Similarly, I would suggest not to go deeper in the diagnostic criteria of canine hypoadrenocorticism since it is not strictly pertinent to the review’ aim and you already mentioned the importance of excluding this endocrine disorder in the diagnostic process of chronic inflammatory enteropathies. I would suggest to remove this part if not strictly necessary, or, at least, cite the latest guidelines for the diagnosis.

Thank you for your advice. According to your advice, I deleted the sentences.

Lines 60-86: This paragraph font is smaller than the remaining manuscript.

Thank you for your kindness. I revised the paragraph.

Lines 60-189: Moreover this paragraph is chaotic and should be rephrased. I would rather suggest to simplify it and provide brief but clear information regarding the diagnostic iter (based on treatment response) in their subsequential steps without citing too many exceptions (i.e Chronic idiopathic colon diarrhea). This review’s aim is not to overview “chronic enteropathies” (or “chronic inflammatory enteropathies”, please eventually check and modify accordingly to which entity you are referring in all the manuscript) but to focus on vitaminic status. Thus, I will reccomend to significantly reduce this introductive part and eventually focus on the mechanism which can cause vitaminic alteration and their potential therapeutic role according to the latest literature.

Line 190: In this point I would suggest to discuss the poor prognostic factors associated with chronic inflammatory enteropathies, before going in the particular of PLE’s ones.

Thank you for your advice. Based on your valuable suggestions, I revised the aim of this review and made some structuring (precise subsections) in introduction.

Line 226: After this part “the most important factor [54]” I would suggest to place a brief explanation about the cobalamin status in canine CE. You directly discuss about the supplementation right after the explanation of the metabolism of cobalamin, but a fundamental junction element is the cobalamin status of these patients.

Thank you for your valuable advice. I added the sentences, as below.

Line 239-240: Indeed, hypocobalaminemia is commonly observed in dogs with CE, and supplementation has been reported to improve clinical signs [62].

Line 238: “sufficient” is an aspecific term, I would suggest to refer to the function without specifying.

Thank you for your advice. I deleted the word.

Lines 242: The font used in Figure 1 and 2 is different, please correct it according to the template guidelines.

Thank you for your kindness. I revised the fonts.

Lines 261: “Due to different diet” is not maybe correct, cause this not a cause rather than the consequence of different Vitamin D metabolism.

Thank you for your advice. I revised the sentence, as below.

Line 277-279: Due to differences in vitamin D metabolism, carnivorous cats cannot synthesize 25(OH)D from ergocalciferol, unlike omnivorous dogs [69, 70].

Lines 274-278: This part is unclear, I would suggest rephrasing to make it easier to understand.

Thank you for your advice. I rephrased and revised sentences, as below.

Lines 293-297: 25(OH)D concentrations are measured when the vitamin D status is ascertained [68]. This is because the blood 25(OH)D concentration is approximately 1000 times easier to measure than the 1,25(OH)2D concentration, and the half-life is as long as 10 days to 3 weeks, which is thought to reflect the actual vitamin D status [71, 72].

Lines 279-284: You state that the hypovitaminosis D in PLE dogs is still in research phase, but I would suggest to consider that this state is well documented in CE dogs; another time, before discussing about its potential prognostic factor, I would rather recommend to talk about the general vitamin D status in dogs with CE (not only PLE), and after, about its potential prognostic role.

Thank you for your valuable advice. Additional sentences have been added for lack of explanation, as below.

Line 294-297: It has been well documented that hypovitaminosis D is often observed in dogs with chronic enteropathy [50, 73-79]. In particular, hypovitaminosis D is more severe in dogs with PLE compared to CE dogs (normoalbuminemia) and is a poor prognostic factor [50, 75].

Lines 296-301: Sorry, I personally can’t understand the role of this paragraph, placed here. It is still discussing about specific therapeutic aspects, not mentioning the vitaminic profile and role.

Thank you for your valuable advice. Additional paragraphs have been added for lack of explanation, as below.

Line 321-326: Digestive enzymes and epithelial barrier function are important for normal digestion [82, 83]. However, it is difficult to assess their function and clinical severity and prognosis. There is also a report that intestinal permeability and mucosal absorptive capacity tests were not useful indicators for estimating clinical disease activity in dogs with CE [84]. Therefore, I focused on nutrition in this manuscript, particularly vitamin B12 and vitamin D, which are poor prognostic factors in dogs with CE and PLE.

Line 327-338: ULFD is not a complete nutritional diet, and its long-term use can cause nutritional problems [44]. Therefore, a complete nutritional diet is the most important aspect of dietary therapy. European Pet Food Industry Federation (FEDIAF) has issued guidelines for complete nutritional diets (https://europeanpetfood.org/wp-content/uploads/2024/09/FEDIAF-Nutritional-Guidelines_2024.pdf). The components of a complete nutritional diet include protein (amino acids; arginine, histidine, isoleucine, leucine, lysine, methionine, cysteine, phenylalanine, tyrosine threonine, tryptophan, valine, taurine), fat (linoleic acid, arachidonic acid, alpha-linolenic acid, eicosapentaenoic acid, docosahexaenoic acid), minerals (calcium, phosphorus, potassium, sodium, chloride, magnesium), trace elements (copper, iodine, iron, manganese, selenium, zinc), and vitamins (vitamin A, vitamin D, vitamin E, vitamin B1, vitamin B2, vitamin B5, vitamin B6, vitamin B12, vitamin B3, vitamin B9, vitamin B7, choline, vitamin K).

Lines 302-308: It is not easy to follow this part and the relationship between intestinal dysbiosis and cobalamin should be more extensively explained, it could be an interesting aspect to discuss.

Thank you for your advice. According to your valuable suggestions, I added the sentences for lack of explanation, as below.

Line 339-348: Serum cobalamin concentrations were negatively correlated with ileal endoscopic and histological severity in dogs with CE [85]. However, the ideal cobalamin receptor was upregulated in dogs with CE [86]. Therefore, cobalamin supplementation is a necessary and reasonable therapeutic strategy, even when administered orally. Recently, oral administration of cobalamin supplementation has been reported to be effective not only for CE dogs but also for PLE dogs [87]. On the other hand, serum MMA concentrations were increased in CE dogs with hypocobalaminemia and have been reported to decrease with cobalamin supplementation [59, 88]. However, increased serum MMA concentrations are sometimes observed even in dogs with normocobalaminemia, and one of the reasons is thought to be involved in intestinal dysbiosis [89].

Line 353-354: These results suggest that intestinal dysbiosis may be associated with increased serum MMA concentrations without hypocobalaminemia.

Line 308: “Similarly, it has been reported that there is no change in dysbiosis index before and after treatment in CE dogs with PLE”, which treatment?

Thank you for your valuable advice. I revised the sentence, as below.

Line 354-356: Similarly, it has been reported that there is no change in dysbiosis index before and after standard treatment with dietary and immunosuppressive therapies in CE dogs with PLE [91].

Line 309-310: “Therefore, fecal microbiota transplantation (FMT) [78] may be a therapeutic strategy for a radical solution to intestinal dysbiosis in the future.”

How this assumption is related to the rest of the paragraph? It seems disconnected and decontextualized. Moreover, it is a strong assumption, I would rather suggest to be more positive and stating that FMT can eventually be considered as a therapeutical tool to modulate the intestinal microbiota (“radical solution” is a strong assumption).

Thank you for your advice. According to your valuable suggestions, I revised the sentence, as below.

Line 356-358: Therefore, fecal microbiota transplantation (FMT) [92] may be a therapeutic tool to modulate intestinal dysbiosis and cobalamin metabolism in the future.

Lines 314-345: “dogs and cats” is repeated twice in the same sentence.

Thank you for your kindness. I revised the sentence.

Lines 321-322: “However, it has not been examined whether this intake is sufficient for 25(OH)D levels in blood”; sufficient to reach what level? This sentence’s meaning is confusing, please rephrase it.

Thank you for your advice. According to your suggestions, I rephrased sentences, as below.

Line 368-369: However, it has not been examined whether this intake provides sufficient 25(OH)D concentrations [68].

Lines 329-331: “Although vitamin D supplementation is not yet clear whether a clear protocol for dosage can be established or whether it improves prognosis, it is certainly a reasonable treatment for PLE dogs with hypocalcemia”

This sentence has issues in its construction which makes it difficult to understand. Moreover, I would remove the word “certainly” since it underscore a certain grade of scientific evidence which you state as not sufficient in the sentence just before this assumption.

Thank you for your advice. According to your suggestions, I added and revised sentences, as below.

Line 376-378: It is reported that there was no statistical difference in duodenal vitamin D receptor expression between healthy dogs and dogs with CE [95]. Therefore, oral vitamin D supplementation may be a therapeutic strategy in dogs with CE. Although vitamin D supplementation is not yet determined a clear protocol for dosage or whether it improves prognosis, it is a reasonable treatment for PLE dogs with hypocalcemia.

Line 333: I would not use the term unknown (see comment above).

Thank you for your advice. I revised the sentences, as below.

Line 382: Canine CE has multiple and different etiologies.

Line 334-335: “Cobalamin and vitamin D are poor prognostic factors in dogs with CE and PLE; therefore, cobalamin supplement in dogs with CE and vitamin D supplement in PLE dogs with hypocalcemia is important.”

I personally believe that this sentence may be conceptually wrong; if a parameter is a poor prognostic factor, it doesn’t necessarily mean that it’s supplementation would be beneficial. Clinical trials are needed to assess whether these supplementations may be useful or not.

I would eventually say that “clinical trials showed that vit.D and B12 supplementations provide benefits to CE dogs”, citing the related studies which can support this assumption.

Lines 338-339: I can’t understand how this assumption may be related to the aim and presumptive main focus of the review (vitamins).

Thank you for your advice. According to your valuable suggestions, I revised sentences, as below.

Line 382-386: Cobalamin and vitamin D are poor prognostic factors in dogs with CE and PLE; therefore, cobalamin supplementation in dogs with CE and vitamin D supplementation in PLE dogs with hypocalcemia may be a reasonable therapeutic strategy. The main treatment strategy in dogs with CE and PLE is complete nutritional dietary therapy.

Line 388-389: Clinical trials are needed to determine whether these vitamin supplementations are useful or not in dogs with CE and PLE in the future.

Round 2

Reviewer 3 Report

Comments and Suggestions for Authors

I appreciate the corrections from the author and I really believe that, regarding the structure and order of stated information, the manuscript has been improved compared to the previous version. Still, I apologize, but I still don't find this manuscprit suitable for the publication. 
Some general comments and some major issues has not been consistently improved: 

Personally, I find this manuscript not suitable for the publication as for how it is presented at the moment.
There are some major issues:
- The review is still mostly focused on chronic enteropathies (classification and general information) while these aspects should just be an introduction, considering the main aim of the review (the role of vitamins D and B12 in canine CE).
- Many assertive assumptions are made from studies with a low “evidence based medicine” grade
- By reading the whole manuscript, the reader is still confused about what is the current literature regarding the vitaminic profile of these kind of patients, which can be the pathophysiological mechanisms involved, what is the clinical evidence of the potential benefits.
- The manuscript still gives confusing information which are not strictly necessary.

Comments on the Quality of English Language

There are many English mistakes and the construction of the sentences sometimes make them difficult to understand.

Author Response

Dear reviewer,

Thank you for your kind and valuable comments.

I submitted my manuscript for English editing.

I corrected my manuscript according to your valuable advice.

The revised part is being shown in red colors.

There are some major issues:

- The review is still mostly focused on chronic enteropathies (classification and general information) while these aspects should just be an introduction, considering the main aim of the review (the role of vitamins D and B12 in canine CE).

Thank you for your advice. Unfortunately, I am unable to make any further major changes because other reviewers have accepted my manuscript.

- Many assertive assumptions are made from studies with a low “evidence based medicine” grade

Thank you for your advice. As you mentioned, there is a paucity of evidence-based literature, especially about vitamin D. Therefore, I have added the following text. Since the title of this Special Issue is “Current Status and Future Challenges of Chronic Enteropathy in Dogs”, I think it seems important to make future evidence sense.

Line 313: However, there are few studies on highly evidence-based veterinary medicine.

- By reading the whole manuscript, the reader is still confused about what is the current literature regarding the vitaminic profile of these kind of patients, which can be the pathophysiological mechanisms involved, what is the clinical evidence of the potential benefits.

- The manuscript still gives confusing information which are not strictly necessary.

Thank you for your advice. Based on your valuable suggestions, I have added text to make it easier also for the readers of paraprofessional to understand.

Line 187-189

Therefore, LFD was added gradually to ULFD after improvement of albumin concentrations in a retrospective study of ULFD treatment in dogs with PLE [43].

Line 240-247

Hypocobalaminemia is commonly observed in dogs with CE [6, 62, 63]. Since cobalamin deficiency of dietary origin has not been reported to date, cobalamin deficiency in dogs with CE may be attributed to the absorption process [54]. Theoretically, there are three main mechanisms that reduce cobalamin utilization in the small intestine: congenital impairment of receptor function, reduced mucosal absorption capacity, and excessive competition with the intestinal microflora [54]. Of these three mechanisms, decreased mucosal absorption capacity due to mucosal inflammation may be the most important in dogs with CE.

Line 249-252

Indeed, cobalamin supplementation has been reported to improve the clinical signs in dogs with CE and hypocobalaminemia [62]. Cobalamin supplementation has been administered subcutaneously to account for the decreased absorption in the intestinal mucosa [54].

Line 314-322

To date, it is not known whether reduced vitamin D concentrations are the cause or only a consequence of intestinal diseases, such as dogs with CE and PLE [68]. A decrease in vitamin D intake or absorption due to the disease may be involved [68]. Indeed, inflammation of the gastrointestinal mucosa inhibits vitamin D absorption, resulting in lower serum vitamin D concentrations [68]. However, there is evidence that low concentrations of 25(OH)D may influence the development of inflammatory bowel process, and 25(OH)D concentrations are negatively correlated with inflammatory mediators such as IL-8 in dogs with CE [68, 77]. Therefore, vitamin D intake, absorption, and intestinal inflammation are closely associated.

Line 352-353

This may explain why cobalamin works orally, even in dogs with CE and severe ileal inflammation.

Line 369-370

Thus, hypocobalaminemia, increased MMA concentration, and intestinal dysbiosis are thought to be a complex combination in dogs with CE.

Line 374-375

ULFD is deficient in vitamin D, which may cause nutritional problems with long-term use [44].

Line 404-415

In humans, the anti-inflammatory role of vitamin D via the vitamin D receptor (VDR) has been well investigated in several diseases, including IBD [95]. Vitamin D inhibits the production of inflammatory cytokines via helper T cells in the gastrointestinal tract, suppresses the growth of harmful intestinal bacteria, and intervenes in intestinal mucosal cells by stimulating proteins involved in membrane junction integrity and intracellular pathogen recognition in human IBD [95]. Moreover, intestinal vitamin D receptor (VDR) expression is significantly decreased in patients with human IBD [96]. However, it has been reported that there was no statistical difference in duodenal VDR expression between healthy dogs and dogs with CE, in contrast to the findings in humans [97]. This may explain why vitamin D works orally, even in dogs with PLE and intestinal inflammation. Moreover, vitamin D supplementation may elicit an anti-inflammatory response through its interaction with VDR.

Round 3

Reviewer 3 Report

Comments and Suggestions for Authors

Dear Author, 
thank you for the deep modifications you applied to your manuscript which in my opinion is now consistently improved. I personally believe that now it may be appropriate for the publication.